# Mathematical Models for the Skin to Lumbosacral Epidural Distance in Dogs: A Cadaveric Computed-Tomography Study

**DOI:** 10.3390/ani11102974

**Published:** 2021-10-15

**Authors:** Tsim Christopher Sun, Mara Schier, Michelle Pui Yan Lau, Fernando Martinez-Taboada

**Affiliations:** Sydney School of Veterinary Science, University of Sydney, Camperdown, NSW 2050, Australia; mara@maraschier.com.au (M.S.); mpylau93@gmail.com (M.P.Y.L.); fer_m_taboada@hotmail.com (F.M.-T.)

**Keywords:** epidural, lumbosacral, mathematical, modelling, locoregional, anaesthesia, computed tomography, ligamentum flavum, epidurography

## Abstract

**Simple Summary:**

Epidural anaesthesia is readily performed in the lumbosacral area in dogs that are undergoing surgery of the hindlimbs or the abdomen. Common techniques that are used to identify the epidural space rely on subtle changes in tactile and audible sensations which are challenging for less experienced clinicians. Research in humans suggest that mathematical equations that are derived from body surface parameters may improve the success of epidural space identification. In a previous study by this research group, we developed two mathematical equations from computed tomography (CT) measurements using dog surface parameters and a body condition score to predict the skin to lumbosacral epidural space. In this study, we aimed to validate the equations in dog cadavers against a gold standard technique (epidurography). For one equation, the use of the occipital-coccygeal length and body condition score resulted in a high degree of correlation and agreement with the skin to lumbosacral epidural space of the cadavers. Future studies will determine whether the knowledge of the skin to lumbosacral epidural space distance prior to needle placement improves the success of epidural space identification.

**Abstract:**

This study aimed to validate previously published computed tomography (CT) derived mathematical equations with the true skin to lumbosacral epidural distance (SLED) in dog cadavers. Phase 1: The lumbar region of 11 dog cadavers were scanned in sternal recumbency to determine the effect of cranial, neutral, and caudal pelvic limb positioning on the CT derived lumbosacral epidural distance (CLED). Phase 2: The epidural space was determined using contrast epidurography, and the SLED was analysed against the mathematical equations using a body condition score (BCS) and either the cadaveric occipital-coccygeal length (OCL) (Equation (1): = 7.3 + 0.05*OCL + 16.45*BCS) or the ilium wing distance (IWD) (Equation (2): = 3.5 + 0.56*IWD + 16.6*BCS). There were no differences detected between the pelvic limb positions and the CLED. Both equations demonstrated strong correlations (Equation (1): *r* = 0.7196; Equation (2): *r* = 0.7590) with the SLED. The level of agreement was greater for Equation (1) than with Equation (2) (concordance coefficient 0.6061 and 0.3752, respectively). Equation (1) also demonstrated a closer fit to the concordance line compared with Equation (2) (bias correction factor 0.8422 and 0.4960, respectively). Further studies in live anaesthetised dogs will help to determine the usefulness of the pre-procedural knowledge when performing lumbosacral epidurals.

## 1. Introduction

Successful epidural anaesthesia relies on the accurate placement of the needle through the skin into the epidural space, but regardless of the method used to identify the epidural space, clinicians are unaware of how far the needle must travel. Most veterinary clinicians performing lumbosacral epidurals do so without any radiographic assistance and instead blindly select the needle trajectory and the final target. Mathematical models have been used in human anaesthesia to predict the distance that is required to enter the epidural space [1,2]. In particular, the relationship between anthropometric variables and the body mass index have been shown to have positive correlations with the skin to lumbar epidural space distance [3,4,5,6,7]. Pre-procedural knowledge of the depth of the epidural space from the skin may serve as a guide for needle placement, particularly for less experienced operators [8,9].

The skin to lumbosacral epidural space distance has recently received similar interest in veterinary anaesthesia [10,11,12]. A recent study in 86 dogs identified significant positive correlations between external variables (occipital-coccygeal length (OCL) and ilium wing distance (IWD)), and the skin to epidural distance using computed tomography (CT) [12]. Multiple regression analyses have also found that the addition of a dog body condition score further improved the relationship between the external variables and the skin to epidural distance. In that study, however, only dogs with their pelvic limbs in caudal and neutral positions were investigated. It is unknown whether cranial pelvic limb positioning, as is commonly performed in dogs that are receiving an epidural, would influence this anatomical relationship. Furthermore, the derived equations from that study were based solely on correlation relationships from measurements that were obtained from the CT images, and the usefulness of the equations in predicting the true skin to lumbosacral epidural distance that are based on external measurements in vivo have yet to be determined. Ideally, a mathematical equation could be used to predict the skin to lumbosacral epidural distance and improve user accuracy in identifying the epidural space before injection in dogs.

The aims of this study were to examine whether the direction of the pelvic limb positioning would affect the skin to lumbosacral epidural distance as determined by CT measurements in canine cadavers. We also aimed to investigate whether external cadaveric measurements would be comparable to the same landmarks on the CT and assess the relationship between the mathematically derived distance and the true skin to lumbosacral epidural distance in canine cadavers. We hypothesised that the pelvic limb positioning would have no effect on the CT derived lumbosacral epidural distance, and that the CT measurements and the cadaveric measurements would be similar. We also hypothesised that the mathematical equations would have close agreement with the true skin to lumbosacral epidural distance.

## 2. Materials and Methods

### 2.1. Cadavers

A prospective experimental study was performed on 11 canine cadavers. The dogs were donated to the Anatomy Department of the School of Veterinary Science for undergraduate veterinary student teaching and were euthanised or died for reasons that were unrelated to the study. The cadavers were thawed in a cold room with a temperature of 18 degrees Celsius for approximately 48 h prior to analysis. The cadavers were fully developed, were not pregnant, and were free of obvious musculoskeletal disease. Any animal that was not amenable to cranial–caudal flexion of the pelvic limbs, suspected to have anatomical defects (i.e., hemi-vertebrae), or had subsequent contamination from the contrast that was outside the epidural space were excluded from analysis. The cadavers were weighed and assigned a body condition score (BCS out of 9 with 9 being obese) [13]. The sex (male or female) and the breed were also recorded.

### 2.2. Procedure

#### 2.2.1. Phase 1

The cadavers were placed in sternal recumbency in the gantry of the CT. In the first phase, three pelvic limb positions were used to determine the effect of positioning on the CT derived skin to lumbosacral epidural distance (CLED) [12], with the pelvic limbs extended caudally (pulled backwards), neutral (placed on the side), and extended cranially (pulled forward). Positioning of the cadavers was performed each time by the same observers (ML and TCS). Scans of the entire lumbar spine and sacrum were obtained using a 16 slice multidetector CT scanner (Phillips 16 Slice, Brilliance CT V2.3; Phillips Medical Systems, Amsterdam, The Netherlands) with a 2 mm transverse slice thickness and 512 × 512 pixel matrix dimensions. For each cadaver, a total of three scans were obtained, one for each pelvic limb position (i.e., lumbosacral epidural cranial, neutral, and caudal positions were LSE_cranial_, LSE_neutral_ and LSE_caudal_, respectively). For each scan, a first straight line was drawn from the cranial margin of the dorsal lamina of the seventh lumbar vertebra to the cranial margin of the first sacral vertebra. This line was defined as the ligamentum flavum surrogate [12]. A secondary line was then drawn to the skin with an angle of 60 degrees to the middle of the first line and was defined as the CLED [12]. Measurements to the nearest millimetre were obtained for each of the three positional scans. The images were analysed by a single observer (TCS) and viewed on three dimensional reconstructions using a bone Hounsfield unit window (Apple Thunderbolt Display, Apple, Cupertino, CA, USA; Mac Mini, Apple; Osirix version 5.7 64-bit, Pixmeo SARL, Bernex, Switzerland).

#### 2.2.2. Phase 2

The second phase of the study investigated whether the surface anatomical measurements in dog cadavers could be used as a replacement of CT measurements and predict the true skin to lumbosacral epidural distance (SLED). Surface measurements of the occipital–coccygeal length (OCL) and ilium wing distance (IWD) were manually measured by the same observer (FMT) to the closest millimetre using a flexible measuring tape with the pelvic limbs remaining in cranial extension (Figure 1). CT measurements were subsequently made from the landmarks of the same cadavers by another observer who was unaware of the initial cadaveric measurements (TCS). The OCL was taken from the occipital protuberance to the median sacral crest of the first sacral vertebrae following the dorsal midline of the cadaver, and the same measurement was performed on the CT image from the scout view. The IWD was the distance measured between the most prominent aspects of the dorsal iliac wings on palpation on the cadaver, and between the dorsal aspects of the ilium wings on the dorsal plane window on the CT where both bony structures were first clearly visible (Figure 1). The measurements were recorded on a computerised spreadsheet for data analysis (Microsoft Excel 2011; Microsoft Corporation, Redmond, WA, USA).

For the determination of the SLED, the pelvic limbs of the cadavers remained in a cranial position and hair from the lumbosacral region was clipped. An experienced veterinary anaesthetist (TCS) performed all of the epidural injections on the cadavers. The running drip method [14] was used initially to enter the epidural space. An 18-gauge 80mm Tuohy epidural needle (B Braun, Melsungen, Germany) was inserted into the lumbosacral space. After the needle was placed through the skin, the stylet was removed and the hub of the needle was connected to a three-way tap (Becton Dickinson, Helsingborg, Sweden) attached to a raised 100 mL saline bag (sodium chloride 0.9%, Baxter International Inc., Sydney, NSW, Australia) using a primed giving set ((Vetquip Niki IV infusion set, Amsino International Inc., Pomona, CA, USA). The needle was advanced until an unequivocal increase in the dripping rate was observed, at which point the three-way tap was removed from the hub of the needle and a scan was performed. The location of the tip of the needle was corroborated by a veterinary radiologist (ML), and if the needle tip was determined not to be in the epidural space, another attempt of the above procedure was performed. Each attempt was defined as a positive running drip followed by a CT scan. Epidurography was then performed in situ using diluted iodinated radiographic contrast (Omnipaque 350; GE Healthcare Australia Pty Ltd., Parramatta, NSW, Australia) with a total of 3 mL solution injected (1 mL contrast and 2 mL saline). Following the contrast injection, a CT scan was performed to confirm the presence of contrast in the epidural space, and the CLED measurement as described above in Phase 1 was later performed with the visible epidural space replacing the first line (LSE_epidurography_) (Figure 2). Once completed, straight forceps were used to clamp the skin–needle interface. A protractor was rested against the skin to measure the acute angle between the Tuohy needle and the skin. The needle and clamp were then removed from the cadaver and the distance from the tip of the needle to the clamp was recorded in mm as the SLED [10]. If epidurography indicated that contrast outside of the epidural space was present, the cadaver was excluded from further analysis.

### 2.3. Statistical Analysis

A sample size of 10 dogs was estimated for the detection of an 8 mm difference [11] between the SLED and two mathematically derived lumbosacral epidural space distances, with a variance of 25, loss of 15% and power of 90% [15]. The mathematical equation for predicting the SLED were obtained by using the BCS with the OCL or IWD derived from our previous study [12]:SLED_equation1_ = 7.3 + 0.05*OCL+ 16.45*BCS(1)
SLED_equation2_ = 3.5 + 0.56*IWD + 16.6*BCS(2)

The BCS was further classified as binominal (0 = less than or equal to BCS 5/9, and 1 = greater than BCS 5/9) as published previously [12]. Measurements were recorded on a computerized spreadsheet (Microsoft Excel 2011; Microsoft Corporation, Redmond, WA, USA), and transferred to statistical software program R, version 3.6.1 for Windows 10 (The R Foundation for Statistical Computing, http: www.Rproject.org, last accessed 18 July 2021, Vienna, Austria).

For Phase 1, the CLED for LSE_cranial_, LSE_neutral_, and LSE_caudal_ were compared. For Phase 2, the cadaveric OCL and IWD were compared with their respective CT measurements, and the LSE_epidurography_ was compared with LSE_cranial_ to validate the ligamentum flavum surrogate. The crude values were examined for normality using the Shapiro–Wilk test. When a normal data distribution was found, a one-way analysis of variance or Student’s t-test was used where appropriate. If the data were non-normally distributed, the Mann–Whitney U test was used. Correlation and concordance relationships were determined to validate the mathematical equations using the cadaveric measurements with the SLED. Correlation was expressed as positive or negative values of Pearson’s correlation coefficient (*r*). Proportion of fit (*R*^2^), F statistic for the predictability variable of the outcome, and β weights for increase in the variable for each standard deviation were reported. Relationships were examined graphically by using the crude values, and the linearity of residuals and fitted values were assessed in R. Concordance was determined by assessing the level of agreement between the predicted variables (Equations (1) and (2)) and a gold standard (SLED) by calculating Lin’s concordance coefficient (*CCC*) and concordance line agreement bias (c.b.), where *CCC* = 1 indicates perfect agreement; 0: no agreement; -1: reverse agreement [16]. Values between 0 and 0.29, 0.3 and 0.49, 0.5 and 0.69, and 0.7 and 1.0 were considered to signify very weak, weak, moderate, and strong for both *r* and *CCC*, respectively [17]. Statistical significance was set at *p* < 0.05.

## 3. Results

### 3.1. Phase 1

A total of 11 cadavers met the inclusion criteria, and no cadavers were excluded. Summary descriptive data for sex, weight, and breed are provided in Table 1. Phase 1 analysis of the pelvic limbs in caudal, neutral, and cranial positions demonstrated no effect on the CLED (*p* = 0.3084).

### 3.2. Phase 2

The individual BCS, SLED, OCL, and IWD (both cadaveric and CT), and the mathematically derived lumbosacral epidural distances using Equations (1) and (2) are shown in Table 1. The cadaveric and CT measurements were different for the IWD (*p* = 0.0150), but not for the OCL (*p* = 0.3).

The contrast epidurography confirmed the SLED in all dogs. Of the dogs, eight required one attempt, two dogs required two attempts and one dog required three attempts at needle placement, while the final angle of the Tuohy needle projection ranged from 60 to 110 degrees. The CLED for the LSE_cranial_ and the LSE_epidurography_ were not different (*p* = 0.6037).

The correlation and concordance of mathematical equations using cadaveric measurements with the SLED are presented in Table 2. Both demonstrated strong correlations with the SLED (Equation (1): *r* = 0.7196, *p* = 0.0125; Equation (2): *r* = 0.7590, *p* = 0.0068). However, the level of agreement was greater for Equation (1) (*CCC* = 0.6061; *p* = 0.0125) than for Equation (2) (*CCC* = 0.3752; *p* = 0.0068). Equation (1) also demonstrated a closer fit to the concordance line (c.b. 0.8422), compared with Equation (2) (c.b. 0.4960).

## 4. Discussion

This study supports the hypothesis that cranial extension of the pelvic limbs in canine cadavers does not affect the CT derived skin to lumbosacral epidural distance when compared with neutral and caudal positions. The cadaveric and CT derived measurements were different for the IWD, but not for the OCL. Furthermore, there was no statistical difference in the CT derived skin to lumbosacral epidural distance when using the ligamentum flavum surrogate and contrast epidurography. Both Equation (1) (using OCL) and Equation (2) (using IWD) correlated with the true skin to lumbosacral epidural space distance, but Equation (2) had higher agreement.

Dogs undergoing clinical epidural placement are frequently placed in sternal recumbency with their pelvic limbs in cranial extension [18,19], and it is possible that the lumbosacral anatomical relationships would be influenced by pelvic limb positioning. Previous studies indicate that dog cadavers with cranially placed pelvic limbs have increased lumbosacral interlaminar distances on the CT when compared with pelvic limbs in the caudal placement [20,21], while the mid-laminar L5-L6 distances in dogs positioned in a sternal kyphotic position with the pelvic limbs extended cranially were greater than in neutral positioning [22]. Our previous research showed that the CT derived skin to lumbosacral epidural distances between pelvic limbs in neutral and caudal positioning were not statistically different [12], and it was unknown whether the pelvic limbs in cranial extension would change this relationship. The results from Phase 1 of this study suggest that cranial extension of the pelvic limbs and the other positions investigated are unlikely to affect the CLED in this population of cadavers. However, there should be caution in extrapolating this relationship to live dogs as the degree of pelvic limb extension may be greater in cadavers than in live dogs. For example, hip extension may be restricted by a disease processes such as osteoarthritis [21]. Additionally, the intrinsic anatomy of the epidural space is maintained by physiological pressures and venous distension, which is lost immediately after death [23]. Further prospective studies in live anaesthetised dogs are necessary to validate the results from this cadaveric study for clinical use.

Epidurography is considered the gold standard method for confirming the epidural space in both dogs and humans [24,25,26], but the use of contrast agents, the cost, and accessibility to advanced imaging limit its use in clinical veterinary practice [27,28]. Epidurography was chosen as the standard for comparison in this study because this method is the most accurate for confirming the epidural placement of a needle [29]. Correct needle placement was essential in identifying the SLED in the cadavers and corroborating the CT derived mathematical equations. In a previous clinical study, the loss-of resistance method was used to determine the depth of the skin to epidural space in anaesthetised dogs [10]. However, this technique has been associated with a procedural failure rate of up to 32% [30] and was considered inadequate in confirming the epidural space when used alone [29]. The current study used the running drip method as a guide prior to epidurography. The running drip method was associated with a success rate of 90% in the detection of the lumbosacral epidural space, but may not differentiate between the intrathecal and epidural spaces [31]. If an alternative technique to epidurography was used as a confirmation for epidural space placement in cadavers, the internal validity of the results may have been reduced.

The results from this study are limited to cadavers and cannot imply any clinical improvement in the detection of the epidural space of dogs in vivo, and the efficacy of performing an epidural was not an aim of this study. Our secondary hypothesis was specific for determining the correlation and agreement between the mathematical equations and the SLED. In humans, predictive mathematical models have been suggested to be beneficial prior to epidural needle placement [6,7,32,33,34]. In particular, mathematical models have been used for people considered to be at risk of a difficult epidural due to deeper distances [9,35,36]. In morbidly obese parturient women, the combination of an ultrasound with pre-procedural epidural depth equations that are determined from height and weight variables improved the success rate of the epidural catheter placement than with ultrasonography alone [9]. However, the equations that were generated from individual studies have not been validated to other patient populations, therefore, their use is limited to within the same study population [8]. Conversely, the present study validated existing equations by demonstrating good agreement and correlation with an alternative unrelated study population [12]. We hypothesised that our previously reported equations using CT derived external morphometric variables would have a high degree of agreement with the SLED using dog cadaveric measurements. However, the CT derived measurement was statistically different to the cadaveric measurements when the IWD was examined. In our study, the dorsal aspects of the iliac crests were difficult to palpate in dogs with a higher BCS. It is possible that the IWD was overestimated in some cadavers due to extra subcutaneous fat cover over the iliac crests, resulting in a wider distribution of datapoints and the lower concordance reported for Equation (2). Alternatively, the measurements of the OCL in both cadaveric and CT derived distances were not different. The cadaveric measurements were performed over two distinct landmarks, the occipital protuberance and the base of the first sacral vertebrae, both of which were easily palpable in all cadavers regardless of the BCS. The practical ease of measuring the OCL and the higher concordance coefficient suggests that using the OCL (Equation (1)) may be a preferred method in predicting the SLED. Further prospective studies in anaesthetised animals can be used to determine whether mathematical equations to predict depth would truly improve the efficacy in epidural space identification.

There were some limitations to this study. Thawed cadavers were used, and the results should not be extrapolated to live dogs. The SLED in cadavers may be greater due to the absence of epidural pressures that are generated by the lymphatic and circulatory systems in the abdominal and thoracic cavities [23]. The effect of freeze-thawing may also have altered the integrity of epidural anatomical structures due to the collapse of the dural sac after freezing [37]. Another limitation was that this study was designed to validate previous mathematical equations that were only formulated for dogs in sternal recumbency, and the results cannot be applied to other recumbencies. A consideration from the results is the clinical perceptible difference that is acceptable between the mathematical equations and the SLED. The sample size calculations suggested that ten dogs were required to detect an 8 mm difference, and while this number may seem small, it was appropriate for demonstrating a meaningful clinical difference [11] and for concordance analysis [16]. The equations used in this study were derived from multivariable analysis involving two external morphometric variables (OCL, IWD), and the BCS. A moderate proportion of fit was found from both Equation (1) (*R*^2^ = 0.5179) and Equation (2) (*R*^2^ = 0.5755), which showed that some variation was unaccounted for in the models. While increasing the number of variables may improve this relationship, it adds complexity and restricts the clinical applicability of these equations. In addition, while fat content in overweight animals is not evenly distributed, the use of the BCS makes the body fat distribution irrelevant, while the binomial classification further reduces subjectivity of multiple BCS categories and improves the practical use of the equations. Some breeds were over-represented in our study population of cadavers; however, the mathematical equations were originally developed incorporating different dog breeds [12], and it would be reasonable to assume that the external morphometric variables and the body condition score would capture some differences in breed conformation.

## 5. Conclusions

In conclusion, cranial extension of the pelvic limbs had no effect on the CT derived skin to epidural distance. Mathematical equations were highly correlated with the SLED and using the OCL showed a higher level of agreement compared with the IWD. Further clinical studies in anaesthetised dogs are required to determine the usefulness of this information prior to epidural needle placement.

## Figures and Tables

**Figure 1 animals-11-02974-f001:**
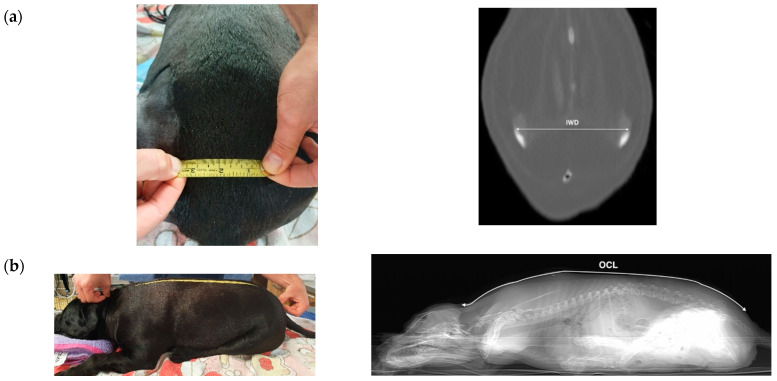
Representative measurements of (**a**) ilium wing distance (IWD), and (**b**) occipital-coccygeal length (OCL) obtained by surface measurements on a dog cadaver (left) and in the same dog on CT (right). Bi-directional arrows depict the landmark sites of the computed tomography (CT) images with the IWD measuring the distance between the dorsal aspects of the ilium wings on the dorsal plane window where both bony structures were first clearly visible, and the OCL from the occipital protuberance to the median sacral crest of the first sacral vertebrae following the dorsal midline of the image from the scout view.

**Figure 2 animals-11-02974-f002:**
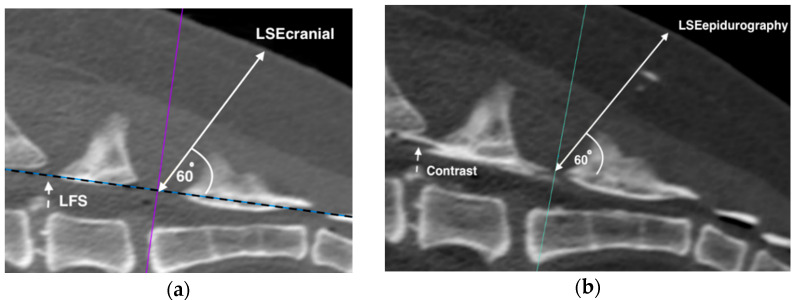
Computed tomography (CT) derived lumbosacral epidural distances (CLED) involving pelvic limbs in cranial extension (LSEcranial (**a**)), and contrast epidurography (LSEepidurography (**b**)). Bi-directional arrows represent the measured CLED distance at 60 degrees from either the ligamentum flavum surrogate (LFS) (hashed arrow (**a**)), or the epidural space confirmed by contrast epidurography (hashed arrow (**b**)).

**Table 1 animals-11-02974-t001:** Descriptive information for the 11 canine cadavers, the CT derived lumbosacral epidural space distance (CLED) based on three pelvic limb positions (Phase 1), with the predictor and the outcome values for correlation and concordance evaluation (Phase 2).

ID	Sex	Weight (kg)	BCS	Breed	Phase 1 (In Millimetres)			Phase 2 (In Millimetres)				
			/9		CLED			OCL	IWD	Equation (1)	Equation (2)	SLED
					Caudal	Neutral	Cranial	Cadaver/CT				
1	M	15	5	Staffordshire bull terrier	43	43	34	565/510	80/59	35.6	48.3	34
2	F	23.5	7	Staffordshire bull terrier	59	53	46	630/593	90/63	55.3	70.5	49
3	F	20.3	6	Staffordshire bull terrier	57	62	52	580/524	95/61	52.8	73.3	51
4	M	29.9	4	Greyhound	36	36	29	950/941	96/102	54.8	57.3	30
5	F	34.1	7	English bulldog	59	63	56	760/687	85/72	61.8	67.7	61
6	M	14.1	4	Beagle	42	41	36	640/609	70/66	39.3	42.7	41
7	M	9.6	2	Crossbred	22	22	17	560/543	60/56	35.3	37.1	24
8	M	14.1	4	Beagle	34	34	28	670/615	80/66	40.8	48.3	31
9	F	9.2	5	French bulldog	33	33	29	455/454	88/53	30.1	52.8	32
10	M	11.2	5	Kelpie	41	40	32	550/527	65/66	34.8	39.9	33
11	M	17.5	4	Staffordshire bull terrier	40	37	30	730/645	90/63	43.8	53.9	29

ID: Cadaver identification; kg: weight in kilograms; Phase 1—the CT derived lumbosacral epidural space measurements of pelvic limbs in the caudal, neutral, and cranial positioning with the cadavers in sternal recumbency; Phase 2: BCS— the body condition score out of 9; OCL: occipital-coccygeal length; IWD: ilium wing distance; Equation (1)—the lumbosacral epidural space derived from the cadaveric OCL using the equation [= 7.3 + 0.05(OCL) + 16.45(BCS)]; Equation (2)—the lumbosacral epidural space derived from the cadaveric IWD using [= 3.5 + 0.56(IWD) + 16.6(BCS)]; SLED: the true skin to lumbosacral epidural distance confirmed with epidurography.

**Table 2 animals-11-02974-t002:** Correlation and concordance relationships between the true skin to lumbosacral epidural distance (SLED), and two mathematically derived lumbosacral epidural distances from 11 canine cadavers.

SLED Prediction	*r*, 95% CI	P*_r_*	*R* ^2^	Intercept	β	F	P_R_^2^	*CCC*, 95% CI	c.b
Equation (1)	0.7196, 0.2107–0.9217	0.0125	0.5179	19.0621	0.6620	9.668	0.0125	0.6061, 0.1471–0.8503	0.8422
Equation (2)	0.7590, 0.2913–0.9340	0.0068	0.5755	22.7683	0.8225	12.2	0.0068	0.3763, 0.0590–0.6246	0.4960

The two mathematical equations: Equation (1) [SLED = 7.3 + 0.05(OCL) + 16.45(BCS)], and Equation (2) [SLED = 3.5 + 0.56(IWD) + 16.6(BCS)], where (BCS) was (0) for body condition score less than or equal to 5/9 and (1) for greater than 5/9, OCL: Occipital-coccygeal length (mm); IWD: ilium wing distance (mm); *r*: Pearson’s correlation coefficient; 95%CI: 95% confidence interval; P*_r_*: *p* value for *r*; *R^2^*: proportion of fit of the variable; F: predictability variable, β: weight for the increase in the variable for each standard deviation; P_R_^2^: *p* value for *R*^2^; β: predictability variability of the outcome; *CCC*: Lin’s concordance coefficient; c.b: agreement bias for deviation from the best fit line of 45 degrees. No deviation occurs when c.b. = 1.

## Data Availability

Data provided on request.

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
