# Peer review of "Mathematical Models for the Skin to Lumbosacral Epidural Distance in Dogs: A Cadaveric Computed-Tomography Study"

_animals, 2021, doi:10.3390/ani11102974_

Round 1

Reviewer 1 Report

The present work is the complement of an earlier study, on the accuracy of the prediction of depth needed to reach the lumbo-sacral epidural space based on two morphological parameters, with the animals in sternal recumbancy with the pelvic limbs cranially positioned. These two parameters are each tied to an equation. Using 10 dogs of various sizes and body conditions, the authors concluded that the occipital-coccygeal length predicts better this depth than the ilium wing distance, and that limb position did not change substantially have an effect.

Overall I find this work to be valid and well designed. It could be slightly perfected, perfecting some arguments, but the core of the work is well described, the statistical analysis is thorough, and the main results are discussed.

I have a few rather small comments however:

The introduction, although sufficient, could be slightly amplified to maybe introduce better the previous work which this manuscript completes. Since only anterior positioning is covered here, it could be useful to contextualize some more.

In the materials and methods, there is an uncertainty in the way the paragraph is written. Although I logically understand that it was, it is not clear in the text whether the contrast injection CT was done using the same epidural needle (and hence reconnecting the needle hub to another bag with the contrast medium) immediately after the previous scan was performed or whether it was a different procedure.

I do not understand the reason why the line of the Ligamentum flavum surrogate is traced from the cranial margin of L7 and not the caudal margin. Is there a specific reason? It seems that, in particular in the case of cranial pelvic limb positioning, the increased angulation between L7 and the sacrum makes it so that the line between the anterior margin of L7 and that of the sacrum would appear much lower.

Discussion:

Regarding BCS and/or weight and/or breed for a relatively small (but provenly sufficient) sample size: could you comment on the value of extrapolation on both ends which (BCS 1 and 9) which are not represented, as well as for very small/large breeds? How confident are you in the equations performance in specific cases?

Similarly, although it is briefly discussed, expanding slightly the explanations regarding the binomial nature of the equations used and how it makes BCS irrelevant is welcome.

In the same sense, since the authors aptly comment on the difference with live animals, a somewhat augmented conclusion/summary of the consequences of the authors findings could help the reader's understanding of the value of the work: Once the right parameter, hence equation, and the limb position factor are considered, how does it all fit together?

Reviewer 2 Report

The paper is well written and clearly explains the procedures undertaken and it deals with the limitations of the work. I would prefer the term 'median sacral crest' is used instead of 'base of the first sacral vertebra' .

Author Response

2) Reviewer two comment and response.

The paper is well written and clearly explains the procedures undertaken and it deals with the limitations of the work. I would prefer the term 'median sacral crest' is used instead of 'base of the first sacral vertebra' .

The authors thank this reviewer for their comments. We have made the alterations.

The OCL was taken from the occipital protuberance to the median sacral crest of the first sacral vertebrae

Reviewer 3 Report

Interesting preliminary study, the bibliography is exhaustive, and the experimental designe clear and well described, The authors well explain the limitations of the study and the need for further investigation for a possible clinical application.

Author Response

The authors thank this reviewer for their comments.